# Automated CT Analysis of Major Forms of Interstitial Lung Disease

**DOI:** 10.3390/jcm9113776

**Published:** 2020-11-23

**Authors:** Marlee S. Crews, Brian J. Bartholmai, Ayodeji Adegunsoye, Justin M. Oldham, Steven M. Montner, Ronald A. Karwoski, Aliya N. Husain, Rekha Vij, Imre Noth, Mary E. Strek, Jonathan H. Chung

**Affiliations:** 1Department of Radiology, The University of Chicago Medicine, Chicago, IL 60637, USA; smontner@radiology.bsd.uchicago.edu (S.M.M.); JChung@radiology.bsd.uchicago.edu (J.H.C.); 2Department of Radiology, Mayo Clinic, Rochester, MN 55902, USA; Bartholmai.Brian@mayo.edu; 3Department of Medicine, The University of Chicago Medicine, Chicago, IL 60637, USA; aadegunsoye@medicine.bsd.uchicago.edu (A.A.); rvij@medicine.bsd.uchicago.edu (R.V.); mstrek@medicine.bsd.uchicago.edu (M.E.S.); 4Department of Medicine, The University of California Davis, Sacramento, CA 95616, USA; joldham@ucdavis.edu; 5Department of Physiology and Biomedical Engineering, Mayo Clinic, Rochester, MN 55902, USA; karwoski.ronald@mayo.edu; 6Department of Pathology, The University of Chicago Medicine, Chicago, IL 60637, USA; aliya.husain@uchospitals.edu; 7Department of Medicine, University of Virginia, Charlottesville, VA 22903, USA; IN2C@hscmail.mcc.virginia.edu

**Keywords:** high-resolution computed tomography, connective tissue disease, hypersensitivity pneumonitis, idiopathic pulmonary fibrosis, interstitial pneumonia with autoimmune features

## Abstract

This study aimed to determine diagnostic and prognostic differences in major forms of interstitial lung disease using quantitative CT imaging. A retrospective study of 225 subjects with a multidisciplinary diagnosis of idiopathic pulmonary fibrosis (IPF), interstitial pneumonia with autoimmune features (IPAF), connective tissue disease (CTD), or chronic hypersensitivity pneumonitis (cHP) was conducted. Non-contrast CT scans were analyzed using the Computer Aided Lung Informatics for Pathology Evaluation and Rating (CALIPER) program. Resulting data were analyzed statistically using ANOVA and Student’s *t*-test. Univariate, multivariable, and receiver operating characteristic analyses were conducted on patient mortality data. CALIPER analysis of axial distribution on CT scans in those with IPF demonstrated greater peripheral volumes of reticulation than either CTD (*p* = 0.033) or cHP (*p* = 0.007). CTD showed lower peripheral ground-glass opacity than IPF (*p* = 0.005) and IPAF (*p* = 0.004). Statistical analysis of zonal distributions revealed reduced lower zone ground-glass opacity in cHP than IPF (*p* = 0.044) or IPAF (*p* = 0.018). Analysis of pulmonary vascular-related structure (VRS) volume by diagnosis indicated greater VRS volume in IPF compared to CTD (*p* = 0.003) and cHP (*p* = 0.003) as well as in IPAF compared to CTD (*p* = 0.007) and cHP (*p* = 0.007). Increased reticulation (*p* = 0.043) and ground glass opacity (*p* = 0.032) were predictive of mortality on univariate analysis. Increased pulmonary VRS volume was predictive of mortality (*p* < 0.001) even after multivariate analysis (*p* = 0.041). Quantitative CT imaging revealed significant differences between ILD diagnoses in specific CT findings in axial and, to a lesser degree, zonal distributions. Increased pulmonary VRS volume seems to be associated with both diagnosis and survival.

## 1. Introduction

The use of high-resolution computed tomography (HRCT) is essential to accurately diagnose interstitial lung diseases (ILDs) in the framework of the multidisciplinary team review [1,2]. Moreover, given recent advances in ILD-specific medical therapy in the past decade, rapid, accurate diagnosis of ILDs has become paramount [3]. Although the HRCT features for diseases such as idiopathic pulmonary fibrosis (IPF) can be diagnostic, the imaging manifestations of different ILDs often have similar appearances. Differentiation of these patterns may be difficult even for those with dedicated training in thoracic radiology [3,4]. This difficulty lends itself to variability and uncertainty despite the existence of diagnostic guidelines. Given the complexity of ILD, diagnostic criteria continue to be refined, especially with regard to HRCT interpretation [5,6,7,8,9].

Quantitative imaging may reveal previously unrecognized diagnostic or prognostic features of ILDs [10]. If such features exist, these analytical tools could facilitate accurate diagnosis, determine prognosis, and refine our understanding of ILDs. It is not yet known what, if any, differentiating features these various forms of ILD might demonstrate when using computer-based automated CT analysis.

The purpose of the current study was to utilize Computer Aided Lung Informatics for Pathology Evaluation and Rating (CALIPER), an analysis tool that enables reproducible quantification, to compare quantitative CT features of various forms of ILD, including IPF, interstitial pneumonia with autoimmune features (IPAF), connective tissue disease (CTD), and chronic hypersensitivity pneumonitis (cHP) [4]. We aimed to specifically examine whether the global, axial (central vs. peripheral), or zonal (upper vs. lower) distributions of specific parenchymal features differed significantly between the four diagnoses. Additionally, we examined the relationship of ILD diagnosis to pulmonary vascular-related structure (VRS) volume, which encompasses the volumes of all pulmonary vessels and their immediately surrounding parenchyma. We also investigated how these variables corresponded with overall mortality. We hypothesized that IPF and IPAF would be characterized by higher global volumes of honeycombing and reticulation, while CTD and cHP would be characterized by higher global volumes of ground-glass opacity [2,5,11,12,13]. We also predicted that axial distributions would demonstrate significant differences in IPF and IPAF compared with CTD and cHP and that zonal distributions would not demonstrate notable differences [9]. Based on CALIPER data in IPF subjects, we also anticipated that pulmonary VRS volumes would be higher in IPF and IPAF than in CTD and cHP. We also hypothesized that pulmonary VRS volumes would be predictive of patient mortality [10]. This study aims to determine diagnostic differences in major forms of ILDs using quantitative CT imaging.

## 2. Experimental Section

### 2.1. Subjects

Subjects were assembled for this retrospective study from our ILD registry. Subjects included all ILD patients in the registry from 2006 to 2015 who had both surgical lung biopsy and CT scans performed at our center within one year of each other, totaling 225 subjects (male: *n* = 129, age, μ = 61.9 years, range = 22.6–84.3; female: *n* = 96, age, μ = 59.7 years, range = 25.1–80; overall: age, μ = 61.0 years, range = 22.6–84.3). This registry was approved by our Institutional Review Board (#14163-A), HIPAA compliance was maintained, and written informed consent was obtained from all subjects included in the study. Chest CT scans from subjects in this study were included as part of previous studies in which qualitative scoring of imaging findings was performed based on visual assessment and semi-quantitative scoring [6,8,9,14]. The current study leverages CALIPER’s quantitative analyses of chest CT in patients with ILD and does not include any qualitative visual assessment. All subjects had a confirmed multidisciplinary diagnosis of ILD, specifically IPF, IPAF, CTD, or cHP. Reaching these multidisciplinary diagnoses involved contributions from pathologists, pulmonologists, rheumatologists, and dedicated chest radiologists according to evidence-based guidelines [15]. Specifically, the diagnosis of chronic cHP was achieved through a multidisciplinary approach as described by the American Thoracic Society [16]. The diagnosis of IPAF was similarly achieved through a multidisciplinary approach. As key literature describing IPAF was first published in 2015, those patients presenting prior to 2015 were retrospectively classified as part of the registry [17]. All included subjects received a non-contrast chest CT scan as part of their clinical care.

### 2.2. Procedures

A sub-millimeter (0.9 mm) axial reconstructed series using a Philips B filter from the earliest non-contrast CT scan available for each subject from our medical center was anonymized and exported from PACS. CT scans were performed on various scanners (Philips Brilliance 16–64-slice scanners or Brilliance iCT 256-slice scanner). A supine CT acquisition was performed through the thorax during end-inspiration at 120 kVp and 220 mAs. CT images were reconstructed using a 512 × 512-pixel image matrix. All scans were then analyzed using the Computer Aided Lung Informatics for Pathology Evaluation and Rating (CALIPER) program, a quantitative analysis tool, and the resulting data were compiled for statistical analysis [4]. CALIPER-generated data included volumetric (mL) parenchymal pattern distributions with classification of each pixel of the lung parenchyma into ground-glass opacity, reticulation, honeycombing, low attenuation areas (such as air trapping and emphysema), or normal tissue [4,18]. The CALIPER program can also classify volumetric findings by their distribution throughout the lung. Global analysis across both lungs, axial analysis of peripheral and central volumes, and zonal analysis of upper and lower lung zone volumes were conducted in this study. The CALIPER software also performs automated segmentation of pulmonary vascular-related structures (VRS), excluding large vessels at the lung hilum [10]. The resulting VRS measurement therefore represents the cumulative integrated cross-sectional area of all vascular structures of all the axial images integrated across slice thickness. CALIPER-generated parenchymal volumes were considered in absolute terms. Relevant clinical data were also collected from the electronic medical record (EMR), specifically patient age and gender.

### 2.3. Statistical Analysis

One-way analysis of variance (ANOVA) tests were used as global tests of significance followed by post-hoc two-tailed Student’s *t*-tests to make pairwise comparisons between means. The ANOVA tests and pairwise comparisons were used to compare specific CT findings across the various diagnoses. This included global volumes of honeycombing/reticulation/ground-glass opacity/low attenuation/pulmonary VRS across IPF, IPAF, CTD, cHP. This also included comparing the peripheral, central, upper zone, and lower zone volumes of honeycombing/reticulation/ground-glass opacity/low attenuation in each lung across the four diagnoses. Both univariate and multivariable analyses were conducted on patient mortality. The univariate analysis, using Student’s *t*-tests, involved comparing patient mortality status to global tissue volumes of specific CT findings such as reticulation as well as pulmonary VRS volume. The multivariable analysis, using a chi-square test, involved the additional variables of age and sex. Receiver operating characteristic analysis was also conducted on the multivariable model. *p* value < 0.05 was considered statistically significant for all tests. All statistical analyses were performed using Wizard Pro software (version 1.9.22, Evan Miller, TN, USA).

## 3. Results

### 3.1. Subject Demographics

Of the 1250 patients in our ILD registry, 225 patients were included in the study. Of these 225 subjects, 25.8% were diagnosed with IPF (*n* = 58), 29.8% with IPAF (*n* = 67), 18.7% with CTD (*n* = 42), and 25.8% (*n* = 58) with cHP. The demographic data of subjects relative to their diagnosis are presented in Table 1. Approximately half of the subjects were ever smokers (58.7%, *n* = 132).

### 3.2. Parenchymal Pattern Distribution

Global analysis of parenchymal pattern volumes by diagnosis revealed no statistically significant differences. Analysis of aggregate ILD features (i.e., summed total volumes of ground-glass opacity, honeycombing, and reticulation across both lungs) by diagnosis also showed no significant differences (see Appendix A, which provides additional data). However, statistical analysis of the axial distribution, specifically the absolute peripheral and central lung volumes, revealed differences in the right lung such that IPF had greater peripheral reticulation than either CTD (*p* = 0.033) or cHP (*p* = 0.007) and IPAF had greater peripheral reticulation than cHP (*p* = 0.041) (ANOVA *p* = 0.026) (Table 2a). In the left lung, CTD had a lower peripheral volume of ground-glass opacity than IPF (*p* = 0.005) and IPAF (*p* = 0.004) (ANOVA *p* = 0.022) (Table 2b). Statistical analysis of zonal distributions revealed reduced left lung lower zone ground-glass opacity in cHP compared to either IPF (*p* = 0.044) or IPAF (*p* = 0.018) (ANOVA *p* = 0.042) (Table 3).

### 3.3. VRS Volume

Global analysis of absolute pulmonary VRS volume by diagnosis indicated greater VRS volume in IPF compared to CTD (*p* = 0.003) and cHP (*p* = 0.003) as well as in IPAF compared to CTD (*p* = 0.007) and cHP (*p* = 0.007) (ANOVA *p* < 0.001) (Table 4).

### 3.4. Mortality in the Context of Diagnosis and Pulmonary VRS Volume

Analysis of multidisciplinary diagnosis and mortality revealed that death was more common in subjects when they had a diagnosis of IPF compared to CTD (*p* < 0.001) and cHP (*p* < 0.00001). Moreover, subjects with a diagnosis of IPAF were more likely to be deceased compared to CTD (*p* < 0.01) or cHP (*p* < 0.00001). Finally, death was more common in subjects with a diagnosis of CTD compared to those with a diagnosis of cHP (*p* = 0.044) (Table 5).

On univariate analysis, increased global volumes of ground-glass opacity (95% CI 747.6–1083.9; *p* = 0.032) and reticulation (95% CI 148.3–229.9; *p* = 0.043) were associated with mortality regardless of diagnosis. Increased pulmonary VRS volume was also strongly associated with patients being deceased, independent of diagnosis (95% CI 191.4–233.0; *p* < 0.001). Total volumes of honeycombing or low attenuation areas were not significant predictors of mortality (Table 6a,b). Similarly, residual normal volumes did not predict death.

Multivariate analysis indicated that mortality was statistically associated with increased age (95% CI 0.011–0.093; *p* = 0.012) and increased VRS volume (95% CI 0–0.015; *p* = 0.041). Mortality was independent of sex and global volumes of ground-glass opacity, honeycombing, reticulation, or air-trapping (Table 7). A corresponding empirical receiver operating characteristic (ROC) curve was constructed, indicating that the multivariate model including tissue volumes (honeycombing, air-trapping, reticulation, ground-glass opacity), age, and sex has predictive ability to discriminate deceased from alive subjects (Area Under Curve (AUC) = 0.745) (Figure 1).

## 4. Discussion

The main findings of this study were that (1) significant differences exist between ILD diagnoses in specific CT findings in axial and, to a lesser degree, zonal distributions; (2) the pulmonary VRS volumes in IPF and IPAF were each statistically greater than those of CTD and cHP; (3) pulmonary VRS volume was an independent predictor of mortality.

Usual interstitial pneumonia (UIP) is the imaging and histological correlate of IPF (Table 8). Given UIP’s high proclivity to involve the subpleural lung, it was not surprising that CT scans of subjects with IPF demonstrated greater peripheral reticulation as compared to CTD or cHP [19]. In some studies, IPAF presents with a UIP pattern, which could explain the greater peripheral reticulation of IPAF compared to cHP [6].

In the context of CTD, nonspecific interstitial pneumonia (NSIP) is the most common form of lung disease, except in patients with rheumatoid arthritis. The majority of cases of NSIP are characterized by a peripherally predominant pattern of ground-glass opacity, amongst other findings [21]. However, a classic finding in NSIP is central lung involvement, which might offer an explanation as to why CTD had lower peripheral volumes of ground-glass opacity compared to other diagnoses [21,22]. One study found peribronchovascular distribution of disease in 37% of NSIP subjects compared to only 5% of UIP subjects [22]. In clinical practice, this is a generally specific finding, which is highly suggestive of NSIP due to a CTD.

The reduced lower zone volumes of ground-glass opacity noted on CT in subjects with cHP compared to IPF was unexpected given that ground-glass is a finding that is commonly associated with cHP rather than IPF. However, this may be explained by the bibasilar predominance of IPF compared to the more diffuse or upper/mid-lung zone distribution of cHP. While extensive ground-glass opacity is suggestive of an alternative diagnosis to IPF, IPF can also feature some amount of ground-glass opacity [23,24]. Guidelines stress the importance of zonal distribution in invoking a non-UIP diagnosis [25]. Our data suggest that the zonal distribution may carry less weight in clinical diagnosis than axial distribution. Indeed, a recent white paper from the Fleischner Society has relaxed its zonal distribution parameters for the typical UIP pattern, now allowing cases with diffuse zonal distribution to be classified as typical UIP when all other findings of UIP are present [20].

Pulmonary VRS in the upper lung zones has been recently identified as a strong predictor of survival in IPF [10]. In the current study, pulmonary VRS volumes in IPF and IPAF were greater than those of CTD and cHP. Moreover, VRS volume was identified as a strong predictor of mortality. Given that ILDs can cause secondary pulmonary hypertension [26,27,28], it might seem reasonable to assume that pulmonary VRS represents an imaging correlate of pulmonary hypertension. However, a previous CALIPER study on IPF suggests that only weak connections exist between VRS volumes, right ventricular systolic pressure, and the carbon monoxide transfer coefficient (Kco) [29]. It has been suggested that pulmonary VRS corresponds to changes in the morphology of pulmonary vascular structures and has been shown to correlate with visual ILD extent [29]. This could potentially be explained by architectural distortion of the vessels due to fibrosis of the surrounding lung.

Increased global volumes of reticulation and ground-glass opacity were associated with mortality. This is consistent with findings that the overall extent of fibrosis, defined as reticulation and honeycombing, in IPF is a strong predictor of mortality [30,31]. Similar results have been found in studies of fibrotic NSIP and UIP [31,32].

Clinically, the results of this study highlight the value of axial distributions in differentiating ILDs. Moreover, the meaning of increased pulmonary VRS volume should be investigated further from a pathophysiological standpoint. In addition, the results highlight the potential prognostic significance of quantitative imaging in ILDs in general. Finally, this study offers insight into the utility of CALIPER as a tool for informing differentiation and general understanding of ILDs.

Our study was limited by its comparatively small number of subjects, although it represents a relatively large study population in the context of other studies of CT imaging in ILDs. This power limitation might contribute to some of our results—notably, the lack of significant associations between global parenchymal pattern volumes and diagnosis. In addition, much of the signal was lost when the data were corrected for CALIPER-derived patient total lung volumes, and the data in our study were thus considered in absolute terms. This signal loss was possibly due to the small parenchymal pattern volumes relative to total lung volumes. While considering the data in absolute terms allows for the observation of more granular findings, it may also permit patient anatomy and size to play a role. The exploratory nature of and large number of variables involved in our study also limited the implementation of a correction for multiple testing in our statistical analysis. Moreover, our study was limited in its retrospective nature. Finally, this study was conducted at a tertiary referral center for ILD management; therefore, the results of this study may not generalize to the community setting.

Regarding generalizability, as CALIPER relies on volumetric histogram analysis and was trained on non-edge-enhanced HRCT data, any alteration in pixel values through edge-enhancing kernels or filters can change the program output. This highlights the need for standardization of image reconstruction techniques across vendors and scanners in order to ensure reliable results. Overall, while severely edge-enhancing algorithms should be avoided when reconstructing CT scans for CALIPER use, most standard scanners and filters should produce compatible scans [2,3].

Our study indicates that, on a cohort-wide scale, IPF demonstrates greater peripheral reticulation than CTD or cHP, IPAF displays greater peripheral reticulation than cHP, and CTD shows lower peripheral volumes of ground-glass opacity than IPF or IPAF. These findings represent significant differences in axial distribution, suggesting that perhaps axial distributions should be strongly considered when making diagnoses between various forms of ILD. Moreover, the implications of pulmonary VRS volume findings should be investigated further in regard to diagnosis as well as survival given growing evidence that VRS has prognostic and diagnostic ramifications.

## Figures and Tables

**Figure 1 jcm-09-03776-f001:**
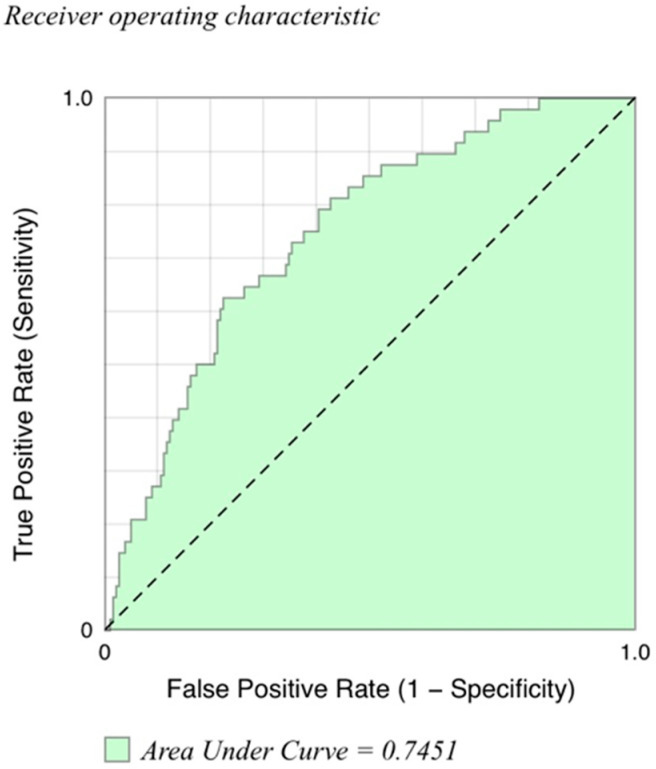
Receiver operating characteristic (ROC) curve for multivariate model for distinguishing deceased from alive subjects. Multivariate model included tissue volumes (honeycombing, air-trapping, reticulation, ground-glass opacity, vascular-related structures), age, and sex. Area under the curve (AUC) for deceased versus alive subjects is 0.745.

**Table 1 jcm-09-03776-t001:** Baseline characteristics stratified by multidisciplinary diagnosis.

Variable	IPF (*n* = 58)	IPAF (*n* = 67)	CTD (*n* = 42)	cHP (*n* = 58)	Total (*n* = 225)
Age (years), mean (±SD)	65.8 (6.9)	60.2 (10.0)	55.0 (12.3)	61.5 (8.6)	61.0 (10.0)
Male gender, *n* (%)	45 (77.6)	33 (49.3)	16 (38.1)	35 (60.3)	129 (57.3)
White race, *n* (%)	54 (93.1)	47 (70.1)	20 (47.6)	48 (82.8)	169 (75.1)
Ever smoker, *n* (%)	41 (70.7)	39 (58.2)	17 (40.5)	35 (60.3)	132 (58.7)

IPF = Idiopathic pulmonary fibrosis; IPAF = Interstitial pneumonia with autoimmune features; CTD = Connective tissue disease; cHP = Chronic hypersensitivity pneumonitis.

**Table 2 jcm-09-03776-t002:** (**a**) Pairwise comparisons between peripheral volumes of reticulation and multidisciplinary diagnosis. (**b**) Pairwise comparisons between peripheral volumes of ground-glass opacity and multidisciplinary diagnosis.

(**a**)					
**CT Finding** **(Range, mL)**	**IPF** **(0.48–280.61)**	**IPAF** **(8.29–317.12)**	**CTD** **(8.34–234.86)**	**cHP** **(1.58–256.21)**	***p*-Value**
Mean reticulation (right lung) (mL)	78.7	73.5			0.565
	78.7		57.7		**0.033**
	78.7			55.4	**0.007**
		73.5	57.7		0.12
		73.5		55.4	**0.041**
			57.7	55.4	0.81
(**b**)					
**CT Finding** **(Range, mL)**	**IPF** **(0.72–686)**	**IPAF** **(0.65–679.18)**	**CTD** **(26.74–549.5)**	**cHP** **(0.28–675.97)**	***p*** **-Value**
Mean ground-glass opacity (left lung) (mL)	264.4	264.8			0.989
	264.4		180.8		**0.005**
	264.4			221.4	0.152
		264.8	180.8		**0.004**
		264.8		221.4	0.14
			180.8	221.4	0.163

IPF = Idiopathic pulmonary fibrosis; IPAF = Interstitial pneumonia with autoimmune features; CTD = Connective tissue disease; cHP = Chronic hypersensitivity pneumonitis. The bolded values represent the significant results.

**Table 3 jcm-09-03776-t003:** Pairwise comparisons between lower zone volumes of ground-glass opacity and multidisciplinary diagnosis.

CT Finding (Range, mL)	IPF (0–362.02)	IPAF (0.32–458.03)	CTD (10.38–435.05)	cHP (0–312.17)	*p*-Value
Mean ground-glass opacity (left lung) (mL)	154.8	162.2			0.678
	154.8		125.2		0.114
	154.8			119.8	**0.044**
		162.2	125.2		0.054
		162.2		119.8	**0.018**
			125.2	119.8	0.767

IPF = Idiopathic pulmonary fibrosis; IPAF = Interstitial pneumonia with autoimmune features; CTD = Connective tissue disease; cHP = Chronic hypersensitivity pneumonitis. The bolded values represent the significant results.

**Table 4 jcm-09-03776-t004:** Pairwise comparisons between absolute pulmonary VRS volumes and multidisciplinary diagnosis.

CT Finding (Range, mL)	IPF (0.79–351.8)	IPAF (68.61–414.99)	CTD (77–420.97)	cHP (2.01–343.03)	*p*-Value
Mean VRS Volume (mL)	195.7	190			0.643
	195.7		153.2		**0.003**
	195.7			156.5	**0.003**
		190	153.2		**0.007**
		190		156.5	**0.007**
			153.2	156.5	0.816

VRS = Vascular-related structures; IPF = Idiopathic pulmonary fibrosis; IPAF = Interstitial pneumonia with autoimmune features; CTD = Connective tissue disease; cHP = Chronic hypersensitivity pneumonitis. The bolded values represent the significant results.

**Table 5 jcm-09-03776-t005:** Patient mortality compared to multidisciplinary diagnosis.

Variable	IPF	IPAF	CTD	cHP	*p*-Value
Fraction deceased	0.379	0.328			0.556
	0.379		0.095		**0.0005**
	0.379			0	**<0.000001**
		0.328	0.095		**0.002**
		0.328		0	**<0.000001**
			0.095	0	**0.044**

IPF = Idiopathic pulmonary fibrosis; IPAF = Interstitial pneumonia with autoimmune features; CTD = Connective tissue disease; cHP = Chronic hypersensitivity pneumonitis. The bolded values represent the significant results.

**Table 6 jcm-09-03776-t006:** (**a**) Univariate analysis relative to alive/dead status: Alive (*n* = 177). (**b**) Univariate analysis relative to alive/dead status: Deceased (*n* = 48).

(**a**)					
**Variable**	**GGO**	**Honeycombing**	**Low Attenuation**	**Reticulation**	**VRS**
Mean (mL)	726.9	12.8	48.7	146.1	166.4
95% CI	649.1–804.6	4.9–20.7	26.3–71.2	127.3–164.8	156.5–176.3
Standard Error	39.4	4	11.4	9.5	5
*p*-value	**0.032**	0.865	0.902	**0.043**	**<0.001**
(**b**)					
**Variable**	**GGO**	**Honeycombing**	**Low Attenuation**	**Reticulation**	**VRS**
Mean (mL)	915.8	11.5	45.5	189.1	212.2
95% CI	747.6–1083.9	4.8–18.1	−9.4–100.4	148.3–229.9	191.4–233.0
Standard Error	83.6	3.3	27.3	20.3	10.3
*p*-value	**0.032**	0.865	0.902	**0.043**	**<0.001**

GGO = ground-glass opacity; VRS = vascular-related structures. The bolded values represent the significant results.

**Table 7 jcm-09-03776-t007:** Multivariable analysis relative to alive/dead status.

Variable	Coefficient	Std. Error	95% CI	Z-Score	*p*-Value
Age	0.052	0.021	0.011–0.093	2.507	**0.012**
Male	−0.264	0.388	−1.025–0.497	−0.681	0.496
Total GGO	0	0	−0.001–0.001	0.071	0.943
Total HC	−0.005	0.006	−0.017–0.007	−0.83	0.407
Total reticulation	0.002	0.002	−0.002–0.005	1.002	0.316
Total VRS	0.008	0.004	0–0.015	2.043	**0.041**
Total low attenuation	0	0.001	−0.002–0.002	−0.176	0.860

GGO = ground-glass opacity; HC = honeycombing; VRS = vascular-related structures. The bolded values represent the significant results.

**Table 8 jcm-09-03776-t008:** Disease-associated CT patterns.

Disease.	Associated CT Pattern(s)	Typical Distribution	Typical Features	Source
IPF	UIP	Basal and subpleural predominant, often heterogeneous	Honeycombing, reticular pattern with traction bronchiectasis	[20]
IPAF	NSIP, OP, LIP, UP	Variable	Variable	[6]
CTD	NSIP	Basal predominant, subpleural sparing	Reticular pattern and ground-glass opacity with traction bronchiectasis	[17]
cHP	HP	Absence of lower zone predominance	Lobular areas with decreased attenuation and vascularity, centrilobular ground-glass nodules	[5]

IPF = Idiopathic pulmonary fibrosis; IPAF = Interstitial pneumonia with autoimmune features; CTD = Connective tissue disease; cHP = Chronic hypersensitivity pneumonitis; UIP = Usual interstitial pneumonia; NSIP = Nonspecific interstitial pneumonia; OP = Organizing pneumonia; LIP = Lymphoid interstitial pneumonia.

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
