# Peer review of "Automated CT Analysis of Major Forms of Interstitial Lung Disease"

_jcm, 2020, doi:10.3390/jcm9113776_

Round 1
Reviewer 1 Report
Specific comments:
Regarding the Experimental section:
Page 2, line 92. Please clarify the timing of diagnosis of IPAF in your patient population. How were patients prior to 2015 diagnosed as IPAF and identified in the registry when the major publication describing this occurred in 2015. Were these classified retrospectively as part of the registry or as part of your specific project? Fischer A, Antoniou KM, Brown KK, Cadranel J, Corte TJ, du Bois RM, Lee JS, Leslie KO, Lynch DA, Matteson EL, et al. An official European Respiratory Society/American Thoracic Society research statement: interstitial pneumonia with autoimmune features. Eur Respir J. 2015;46(4):976–87.
Page 2, line 94 – Does your cohort include both subacute and chronic HP or is it only chronic HP? If both, please clarify this in the text to specifically mention including both as well as the percentages of each in the cohort. If possible, would be great to see them listed as fibrotic/nonfibrotic given the updated guidelines but I do not believe this is required for further consideration if not feasible since your cohort is from pre 2015. If all patients are chronic HP or fibrotic, please change the other references to HP in the manuscript to cHP or fHP to make it clear to the reader that this work does not include evaluation of the subacute/nonfibrotic patterns of HP.
Regarding the Results:
Well presented. No questions
Regarding the Discussion:
Well discussed.
Please also discuss in more detail why signal is lost when correcting data for CALIPER total lung volumes. If this is previously described, then reference that text.
Please also mention any potential generalizability issues that could arise in this type of quantitative analysis should a different reconstruction or different vendor be used for the CT acquisitions. Example, if a facility uses Siemens hardware and a B60 filter.
Regarding the figures:
No changes
Reviewer 2 Report
The authors set out to analyze the distributions of pulmonary abnormalities, using the well published automated CT analysis tool CALIPER, among a moderate sized population of patients who have been diagnosed with IPF, IPAF, CTD, and HP from a multidisciplinary interstitial lung disease board at one tertiary academic center. Of the 225 total patients analyzed, 58 were with IPF, 67 were with IPAF, 42 were with CTD, and 58 were with HP, the more common categories of interstitial lung diseases.
Many of the findings were not surprising, such as IPF being associated with more peripheral reticulation over CTD and HP, and HP showing lesser lower zone GGO then IPF or IPAF. A few other findings are somewhat unexpected, such as CTD being less associated with lesser amounts of peripheral ground glass than IPF and IPAF, and raise questions that question the standard descriptors of CTD, and raises direction for further research. This study interestingly suggests that axial distribution may be more important than craniocaudal distribution in assessment of ILD.
Limitations:
This is a retrospective analysis of association with disease features as measured by a quantitative analysis program. There is a little bit of a circular logic, self-fulfilling data in the study, as that the original radiologist interpretation contributed to the multidisciplinary board review of cases, and imaging findings befitting the described categories contribute to the classification of ILD into IPF, IPAF, CTD, and HP. Nevertheless, this is not a fatal flaw, as that pathology is not the final diagnosis in cases of ILD, and Multidisciplinary conference is about as “gold standard” as possible in a real world clinical scenario.
It is interesting and commended that the authors point out that much of the signal observed was lost when data was corrected for caliper derived patient total lung volumes, as it seems that the opposite might be true. This might be direction for more investigation.
Utility:
Moving from qualitative and often subjective assessments of CT interpretation to more standardized quantitative approaches are important for advancement of lung imaging. This work helps to advance this perspective.
Given the high degrees of overlap of lung abnormalities among the different categories, it is unlikely that quantitative analysis will be the end-all in ILD CT pattern classification. Although it is helpful to see caliper enable a previously unattainable measure of vascular related structures (VRS) to be rapidly quantified, and again demonstrate in this paper that VRS is associated with greater likelihood of death. Further studies are needed to apply the predictors of a poor outcome to a larger population, but quantitative CT is promising in allowing for some degree of prognostication based on imaging features.
Suggestions:
Provide graphic examples of caliper analysis of the four major categories with stereotypical findings. Such as a figure with coronal views of CTs of IPF, IPAF, CTD, and HP with the matching color coded caliper classifications, and glyphs.
Tables 6a and 6b. Unclear as to appropriateness of “air trapping” category given that the CT analyze were only performed in single inspiratory phase. CALIPER classifies low density areas. Given that the lack of expiratory phase CT, it’s not clear that low density areas translate to air trapping. Given that high percentage of the patients are or were smokers (more than half) emphysema is certainly possible, and the “air trapping” column probably needs to be rephrased to something else, like “low attenuation.”
Minor editing point
Page 2, line 67 remove the word “is” in “which is encompasses…”
